# Cryo-EM structures of S-OPA1 reveal its interactions with membrane and changes upon nucleotide binding

Danyang Zhang[1,2†], Yan Zhang[1†], Jun Ma[1,2], Chunmei Zhu[1,2], Tongxin Niu[3], Wenbo Chen[1,2], Xiaoyun Pang[1], Yujia Zhai[1], Fei Sun[1,2,3]*

[1]National Key Laboratory of Biomacromolecules, CAS Center for Excellence in Biomacromolecules, Institute of Biophysics, Chinese Academy of Sciences, Beijing, China; [2]University of Chinese Academy of Sciences, Beijing, China; [3]Center for Biological Imaging, Institute of Biophysics, Chinese Academy of Sciences, Beijing, China

**Abstract** Mammalian mitochondrial inner membrane fusion is mediated by optic atrophy 1 (OPA1). Under physiological conditions, OPA1 undergoes proteolytic processing to form a membrane-anchored long isoform (L-OPA1) and a soluble short isoform (S-OPA1). A combination of L-OPA1 and S-OPA1 is essential for efficient membrane fusion; however, the relevant mechanism is not well understood. In this study, we investigate the cryo-electron microscopic structures of S-OPA1–coated liposomes in nucleotide-free and GTPγS-bound states. S-OPA1 exhibits a general dynamin-like structure and can assemble onto membranes in a helical array with a dimer building block. We reveal that hydrophobic residues in its extended membrane-binding domain are critical for its tubulation activity. The binding of GTPγS triggers a conformational change and results in a rearrangement of the helical lattice and tube expansion similar to that of S-Mgm1. These observations indicate that S-OPA1 adopts a dynamin-like power stroke membrane remodeling mechanism during mitochondrial inner membrane fusion.

*For correspondence:
feisun@ibp.ac.cn

†These authors contributed equally to this work

Competing interests: The authors declare that no competing interests exist.

## Introduction

In eukaryotic cells, series of discrete membranous compartments separate different biochemical reactions, and the membrane fission and fusion mechanisms accomplish communication between and within these compartments (*McNew et al., 2013*). A family of large GTPases, called dynamins, are instrumental in fission and fusion processes (*Praefcke and McMahon, 2004*). Mitochondria, which are double-membrane organelles, can form remarkable dynamic networks through membrane fusion and fission regulated by dynamins (*Labbé et al., 2014*; *van der Bliek et al., 2013*; *Westermann, 2010*). Among those dynamins, optic atrophy 1 (OPA1) is known to be related to mitochondrial inner membrane fusion (*Anand et al., 2014*; *Frezza et al., 2006*; *MacVicar and Langer, 2016*).

OPA1 contains an N-terminal mitochondrial targeting sequence (MTS), a following transmembrane domain (TM), a coiled-coil domain, a highly conserved GTPase domain, a middle domain, and a C-terminal GTPase effector domain (GED). OPA1 has eight spliced variations in the region between the TM and the coiled-coil domain (*Belenguer and Pellegrini, 2013*; *Olichon et al., 2007*; *Figure 1A*). After being imported into the mitochondria, the MTS is proteolytically processed with the remaining parts to form a membrane-anchored long-form OPA1 (L-OPA1). L-OPA1 can be further cleaved into a soluble short-form OPA1 (S-OPA1) through the S1 or S2 site between the TM and the coiled-coil domain (*Ishihara et al., 2006*; *Song et al., 2007*). Both L-OPA1 and S-OPA1 participate in mitochondrial inner membrane fusion. However, the specific role of S-OPA1 during the fusion process remains unclear (*Anand et al., 2014*; *Ban et al., 2017*; *Del Dotto et al., 2017*). The

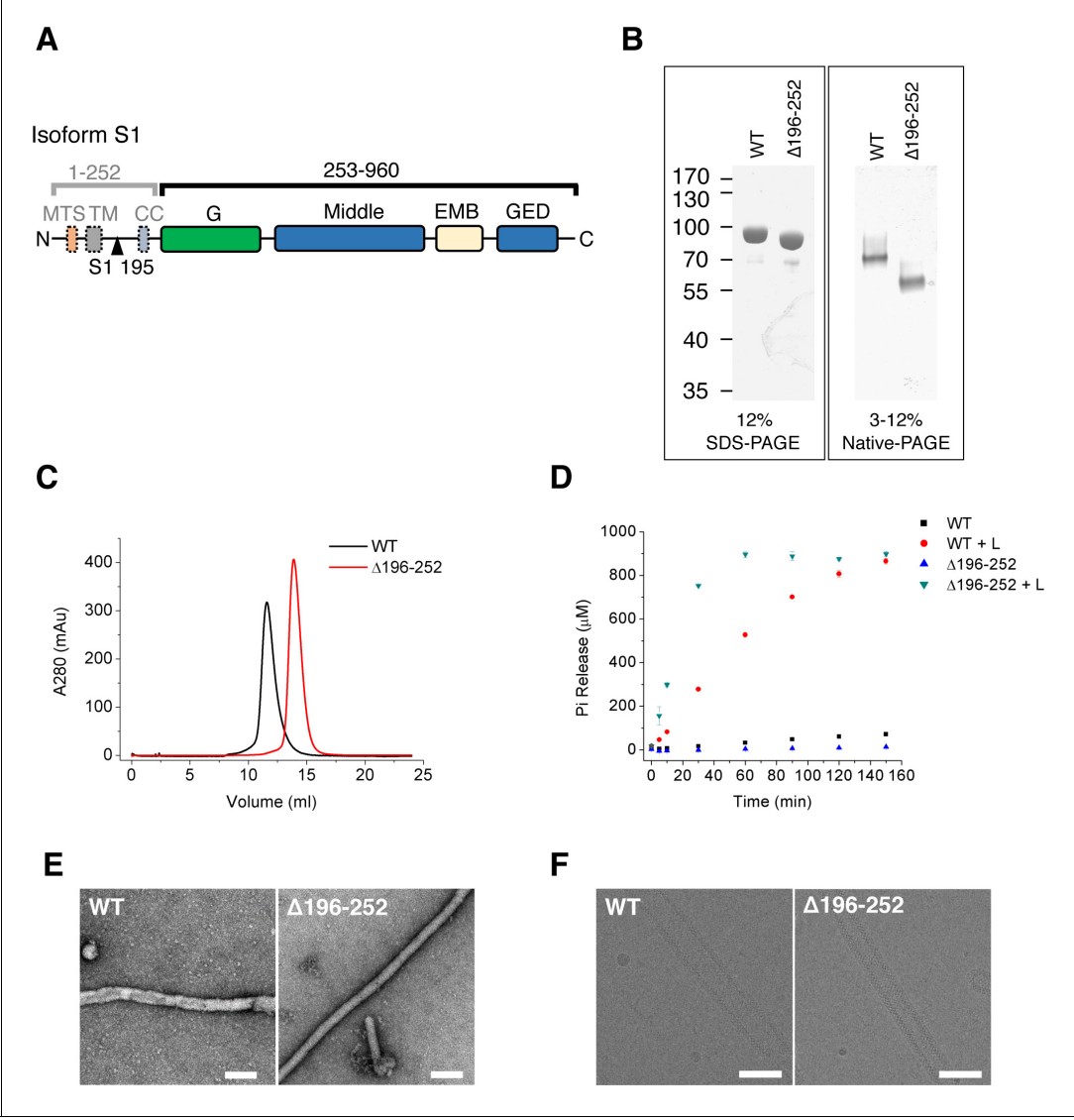

**Figure 1.** Purification and characterization of S-OPA1. (A) Domain organization of OPA1. MTS, mitochondrial targeting sequence; TM, transmembrane region; CC, coiled-coil; G, G domain; Middle, middle domain; EMB, extended membrane binding domain; GED, GTPase effector domain. The proteolytic cleavage site S1 in isoform 1 at the 195th residue is indicated by black triangle. (B) SDS-PAGE and native PAGE of wild type S-OPA1 and its truncation form (Δ196-252). (C) Size exclusion chromatography of S-OPA1 using Superdex 200 10/300 GL column (GE Healthcare). (D) Basal and liposome-binding induced GTPase activity of S-OPA1 and Δ196-252. The total free phosphate was measured at each time point after adding GTP to 1 mM and data presented come from 3 independent experiments. See also *Source data 1*. (E) Negative stain electron micrographs of S-OPA1 coated tubes. Scale bar, 200 nm. (F) Cryo electron micrographs of S-OPA1 coated tubes. Scale bar, 100 nm.

The online version of this article includes the following figure supplement(s) for figure 1:

**Figure supplement 1.** Characterization of S-OPA1.

results of efforts to recapitulate the fusion mechanism in vitro by using Forster resonance energy transfer have indicated that L-OPA1 alone on either side of the membrane can promote fusion with an appropriate concentration of cardiolipin on the opposite side (*Ban et al., 2017*). By contrast, S-OPA1 forms a bridge to the opposite membrane, probably through interactions with both L-OPA1 and cardiolipin, and then assists in L-OPA1–dependent fusion, which involves higher GTPase activity (*Ban et al., 2017*). Studies on Mgm1, the yeast homolog of OPA1, have similarly concluded that its long form, L-Mgm1, acts as a fusion-prone protein with inhibited GTPase activity while its short-form, S-Mgm1, drives the fusion process through GTP hydrolysis (*DeVay et al., 2009*; *Zick et al., 2009*). Another study of S-OPA1 confirmed its tubulation activity with cardiolipin-containing

liposomes by using negative-staining electron microscopy (nsEM) (*Ban et al., 2010*). These studies suggested a GTPase-dependent auxiliary function of S-OPA1 during membrane fusion. However, another report also supported the favorable function of S-OPA1 in fission because it was observed co-localizing in mitochondrial constriction sites (*Anand et al., 2014*).

A 2019 analysis of the structures of S-Mgm1 in *Chaetomium thermophilum* (*Faelber et al., 2019*) indicated a classic dynamin-like structure of S-Mgm1 with only two interfaces during oligomerization on the membrane. S-Mgm1 probably adopts a GTP-dependent power stroke similar to that of dynamin 1 (Dyn1) and deforms the membrane in different ways while binding to negatively and positively curved membranes. In contrast to Dyn1, the binding of nucleotides may cause the expansion of an S-Mgm1-coated liposomal tube because S-Mgm1 has a left-handed assembly geometry. Another crystal structure of *Saccharomyces cerevisiae* S-Mgm1 may provide another model of membrane deformation through the trimeric structure of S-Mgm1 (*Yan et al., 2020*). These results elucidate how S-Mgm1 oligomerization contributes to mitochondrial inner membrane fusion and cristae biogenesis. To determine whether S-OPA1 employs similar methods and further understand the mechanism of mitochondrial inner membrane fusion, we conducted biochemical studies of S-OPA1 and the cryo-electron microscopy (cryo-EM) structures of S-OPA1–coated liposome tubes in a nucleotide-free state and a GTPγS-binding state. Our study provides further molecular insight into mitochondrial inner membrane remodeling.

## Results

### S-OPA1 can induce tubulation of cardiolipin-containing liposomes

We expressed splice form 1 human S-OPA1 (*Figure 1A*) in bacteria and purified it until homogeneity was achieved (*Figure 1B*). The results of gel filtration and chemical cross-linking experiments indicated the presence of a dimerization form of S-OPA1 in the solution (*Figure 1C*; *Figure 1—figure supplement 1*). The GTP hydrolysis activity of S-OPA1 was weak but remarkably enhanced by approximately 70-fold (Kcat) in the presence of liposomes (*Figure 1D*; *Supplementary file 1*; *Source data 1*). The liposomes were prepared with a phospholipid composition of 45% 1,2-dioleoyl-sn-glycero-3-phosphocholine, 22% 1,2-dioleoyl-sn-glycero-3-phosphoethanolamine, 8% phosphatidylinositol and 25% cardiolipin, which approximately reflects the composition of the mitochondrial inner membrane (*Ban et al., 2010*). By examining the mixture of S-OPA1 (apo) with a liposome using both nsEM and cryo-EM, we identified considerable tubulation of the liposomes induced by S-OPA1 (*Figure 1E and F*). However, the tubes varied in diameter, suggesting the dynamic behavior of S-OPA1 on the membrane (*Figure 2—figure supplement 1A*).

To improve the homogeneity of S-OPA1–coated tubes, we purified a truncated form of S-OPA1 (Δ196-252, see *Figure 1A and B*) by deleting its N-terminal dimerization-inducing coiled-coil region from 196 to 252 (*Akepati et al., 2008*). In contrast to its full-length wild type (WT), the truncated S-OPA1 behaved as a monomer in gel filtration and chemical cross-linking (*Figure 1C*; *Figure 1—figure supplement 1B*). Its basal GTP hydrolysis activity was slightly lower than that in the WT, but the addition of liposomes could lead to a more than 100-fold increase (*Figure 1D*; *Supplementary file 1*; *Source data 1*). In addition, the truncation could also induce liposome tubulation but with greater homogeneity and a smaller diameter (*Figure 1E and F* and *Figure 2—figure supplement 1A*). Thus, we selected Δ196-252 for the subsequent cryo-EM structure studies. However, for all the subsequent biochemical and biophysical assays, the full-length WT was used.

### Helical structure of S-OPA1–coated tube in a nucleotide-free state

To characterize the structure of nucleotide-free S-OPA1–coated tube, we collected a cryo-EM data set and classified the boxed tubes according to diameter and diffraction pattern (*Figure 2—figure supplement 1A and B*). A selected class of tubes with an average diameter of 53 nm were segmented and further reconstructed using the iterative helical real-space reconstruction (IHRSR) algorithm (*Egelman, 2000*; *Egelman, 2007*; *Figure 2—figure supplement 1C and D*). This approach yielded a six-start left-handed helical map with a resolution of approximately 15 Å. The tube had an inner diameter of 23 nm, an outer diameter of 53 nm, 17.3 units per turn, and a pitch of 465.6 Å (*Figure 2A and B*; *Video 1*).

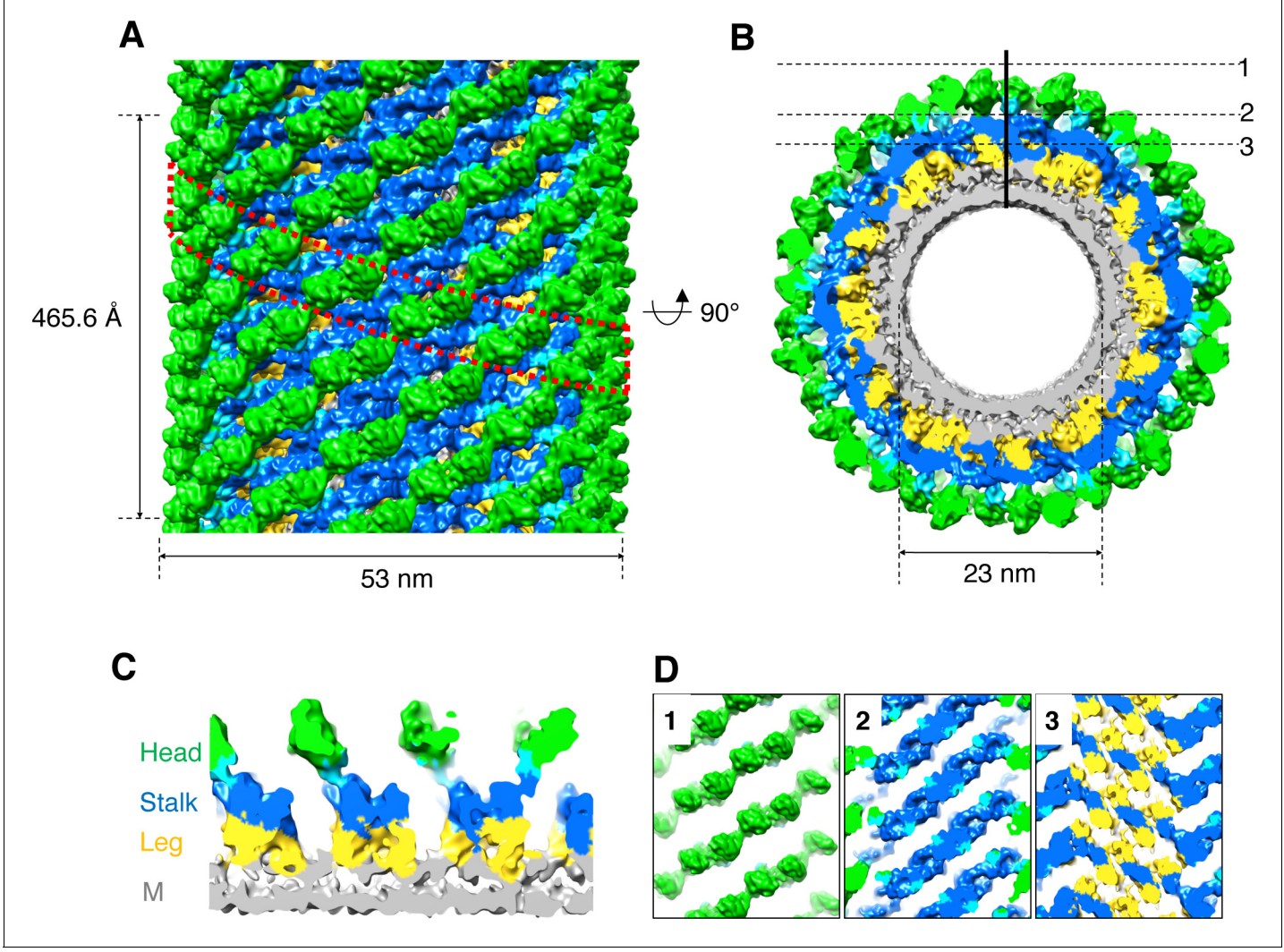

**Figure 2.** 3D reconstruction of nucleotide-free S-OPA1 coated tube. (A) Side view of cryo-EM map of S-OPA1 coated tube. Other than membrane, the map is subdivided and colored radially into three layers denoting ''leg'' (yellow), ''stalk'' (blue), and ''head'' (green and cyan). The outer diameter and pitch are labeled. A single helical rung is highlighted in red dashed box. (B) Radical cross-section of the tube. The inner diameter is labeled. Dashed black lines denote the planar sections that are rotated by 90° and shown in (D). (C) Cross-section of the tube along the solid vertical black line in (B). The leg, stalk, head, and membrane bilayer density are labeled and colored as in (A). (D) Corresponding cross sections of the tube along the dashed black lines in (B). The density color scheme is same as in (A).

The online version of this article includes the following figure supplement(s) for figure 2:

**Figure supplement 1.** Image processing of S-OPA1 coated tubes.

**Figure supplement 2.** Sub-tomogram averaging of S-OPA1 coated tubes at nucleotide-free state.

Similar to Dyn1 coated tube (*Chappie et al., 2011*; *Sundborger et al., 2014*), the S-OPA1 coated tube can generally be divided into three regions along its radial direction: the inner layer is fused with the outer leaflet of the lipid bilayer, the middle part has a stick-like density with compact packing, and the outer layer contains separated globular blocks (*Figure 2C and D*). We labeled the inner, middle, and outer densities leg, stalk, and head, respectively, according to the nomenclature of Dyn1 coated tube (*Chappie et al., 2011*).

Notably, because we could not confirm the handedness of S-OPA1 coated tube at the resolution provided after helical reconstruction, we further performed cryo-electron tomography (cryo-ET) together with subvolume averaging (SVA) on the same sample (the handedness of this procedure was precalibrated). The final averaged cryo-ET map revealed a consistent architecture compared with that of the helical reconstruction (*Figure 2A*; *Figure 2—figure supplement 2A*). To eliminate

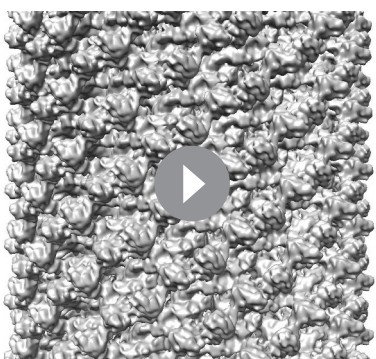

**Video 1.** Structure of S-OPA1 coated tube at nucleotide-free state.
https://elifesciences.org/articles/50294#video1

possible artifacts resulting from the truncation, we also determined the structure of full-length S-OPA1 coated tube by using the same tomographic procedure (*Figure 2—figure supplement 2B*). The result indicated that it had a similar density architecture to that of the truncated form. Thus, the S-OPA1 helical lattice that we observed to be bound to the membrane was not an artifact of the truncation of the coiled-coil domain.

## Domain organization of S-OPA1 and membrane-binding sites

Dynamin proteins have a similar domain architecture (*Figure 1A*; *Figure 3—figure supplement 1*), and structure predictions made using Phyre2 (*Kelley et al., 2015*) and Iterative Threading ASSEmbly Refinement (I-TASSER) (*Roy et al., 2010*; *Yang et al., 2015*; *Zhang, 2008*) revealed that S-OPA1 has a classic Dyn1-like general structure (*Figure 3—figure supplement 2A*). Sequence alignments also indicated that OPA1 and Mgm1 have the highest similarity (42.4%) among the dynamin proteins (*Figure 3—figure supplement 1*). Thus, we docked the crystal structure of *C. thermophilum* S-Mgm1 (PDB ID 6QL4) (*Faelber et al., 2019*) into our cryo-EM map. Considering the possibility that relevant conformational changes would occur in different domains, we separated the crystal structure into three parts: the G/BSE region, the middle/GED stalk, and the paddle domain (*Figure 3—figure supplement 2A*).

The crystal structure of S-Mgm1 fits well into both the helical reconstruction and subtomogram averaged maps, except for a slight bend in the paddle domain (*Figure 3A-D*; *Figure 2—figure supplement 2*). The G/BSE region could be well fitted into the head layer and the following linker that connected the head and stalk layers. The adequate fit suggests that S-OPA1 has a similar G domain structural component (GTPase domain) and BSE three-helix bundle to that of S-Mgm1. Because the G domains of dynamin proteins require dimerization to activate GTPase activity (*Chappie et al., 2010*; *Gasper et al., 2009*), we also investigated whether the G domains in S-OPA1 are dimerized in the present cryo-EM density. However, attempts to dock the crystal structure of dimerized G domains $GG_{GDP.AlF4}$- (PDB ID 2X2E) (*Chappie et al., 2010*) failed (*Figure 3—figure supplement 3A*). Furthermore, although the dimerization interfaces of the proximal G domains in the packing faced each other, they were still separated by approximately 40 Å (*Figure 3D*). This suggests a further conformational change after the subsequent nucleotide binding.

The stalk region of S-Mgm1 could also be well docked into the stick-like middle density layer of the map (*Figure 3A-D*). The whole stalk region of the S-OPA1 array exhibits compact packing with three interaction interfaces (*Figure 3D*), namely the side-to-side (I1), tip-to-tip (I2), and center-to-center (I3) interfaces. Such compact packing suggests that stalk interactions play a pivotal role in maintaining the structural stability of the S-OPA1-lipid complex.

The leg density of the S-OPA1 tube could accommodate the S-Mgm1 paddle domain (*Figure 3B-D*). Sequence analysis did not clearly indicate a paddle-like membrane-binding domain existing in S-OPA1 (*Figure 3—figure supplement 1*). However, the S-Mgm1 paddle domain fit well into the inner density (*Figure 3D*), suggesting S-OPA1 has an extended membrane-binding (EMB) domain of a comparable size, and this EMB domain interacts with the mitochondrial inner membrane. The corresponding sequence of such a domain would be located between the middle and GED domain of S-OPA1 (*Figure 1A*; *Figure 3—figure supplement 1*). The EMB domains of S-OPA1 interact with each other and contribute to another interface (interface P1, *Figure 3D*) for S-OPA1 assembly on the membrane. Notably, the sequence in the EMB domain did not exhibit high conservation of S-Mgm1 (*Figure 3—figure supplement 1*), which might explain the extra unfitted density underneath stalk interface-2 (*Figure 3D*).

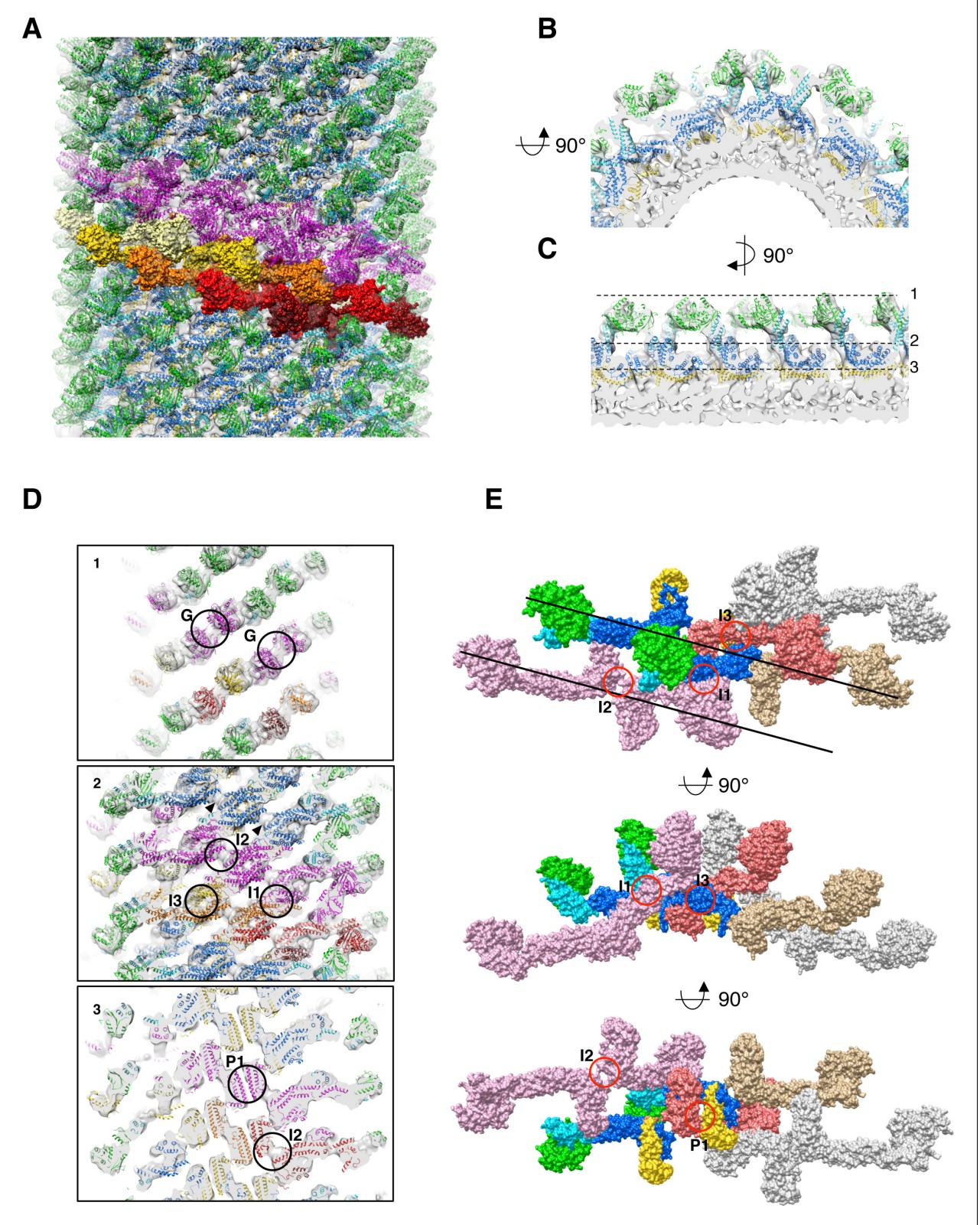

**Figure 3.** Docking of S-Mgm1 crystal structure into cryo-EM map of S-OPA1 coated tube at nucleotide-free state. (**A**) Docking crystal structure into helical reconstructed cryo-EM map (transparent gray). G domain is colored in green, BSE domain in cyan, middle/GED stalk in blue, and paddle domain in yellow. The magenta colored models represent molecules in one helical rung of S-OPA1 assembly. And the building blocks of one helical rung, the dimers of S-OPA1, are shown in the surface representation with the color dark-red, red, gold, yellow and light-yellow, respectively. (**B**)
*Figure 3 continued on next page*

*Figure 3 continued*

Zoomed–in view of radical cross-section showing the fitness between structural model and the map. (C) Vertical cross section of the map that rotates 90° with respect to (B). Dashed black lines denote the positions of the cross sections in (D). (D) Cross sections of the tube showing the fitness between structural model and the map. Putative G dimer interface (G) is shown in panel 1. The stalk interaction interfaces-1,2 and 3 (I1, I2 and I3) are indicated in panel 2. And the paddle interaction interface P1 as well as interface-2 (I2) are indicated in panel 3. (E) Structural model of the left-handed S-OPA1 assembly on membrane at nucleotide-free state. The four interfaces (I1, I2, I3 and P1) for the stability of S-OPA1 assembly are indicated with red circles. The orientation of the helical rung is indicated with black lines.

The online version of this article includes the following figure supplement(s) for figure 3:

**Figure supplement 1.** Sequence alignment of *C. thermophilum* Mgm1 (CtMgm1), S. cerevisiae Mgm1 (ScMgm1) and human OPA1 isoform 1 (HsOPA1).

**Figure supplement 2.** Structural model of S-OPA1 and its helical assembly.

**Figure supplement 3.** G domain dimerization interface analysis.

**Figure supplement 4.** Interfaces comparison between different dynamin proteins.

**Figure supplement 5.** Potential G dimers in different nucleotide binding states.

## S-OPA1 dimer is the building block of its helical assembly on the membrane

The helical reconstruction processing procedure indicated that S-OPA1–coated tube contains six helical starts. By fitting S-Mgm1 crystal structures into the map, we observed that the asymmetric unit of the S-OPA1 packing array contains two S-OPA1 molecules (*Figure 3A*). They form a dimer through the tip-to-tip interface (interface-2, I2, *Figure 3D and E*) similar to the interface-2 of Dyn1 and S-Mgm1 dimers (*Figure 3—figure supplement 4*; *Faelber et al., 2019*; *Reubold et al., 2015*). This interface remains at the distal position of the stalk and at the connection site between the stalk and the EMB domains. Density at this interface could not be well fitted by the S-Mgm1 crystal (*Figure 3D*). This could be attributed to the structural difference between S-OPA1 and S-Mgm1, which is consistent with the sequence variation at the stalk and the EMB/paddle domains (*Figure 3—figure supplement 1*).

We then investigated how the S-OPA1 dimers assemble on the liposomal tube (*Figure 3*; *Figure 3—figure supplement 2*). By following the left-handed helical symmetry, S-OPA1 dimers interacted with each other through stalk interface-3 and the EMB domain interface P1. The packing array was further stabilized by interface-1 among different rungs (*Figure 3E*; *Figure 3—figure supplement 2B and C*). Interface-1 in the S-OPA1 oligomer corresponds to the interface 1 in Dyn1 and S-Mgm1 tetramers, whereas the interface-3 and P1 are unique to S-OPA1. In addition, the S-OPA1 G domains within one single helical rung were facing each other with their dimerization interfaces, which leaves the possibility of a subsequent change of the S-OPA1 array (*Figure 3D*; *Figure 3—figure supplement 5*).

However, when considering the helical assembly of S-Mgm1, which uses interface-1 and interface-2 to form a helical rung, we could also model the S-OPA1 dimer assembly in a right-handed manner (*Figure 3—figure supplement 2D and E*), where interface-3 and interface P1 stabilize the interaction between neighboring helical rungs. Moreover, in the right-handed assembly, the S-OPA1 G domains from neighboring rungs were facing each other with their dimerization interfaces (*Figure 3—figure supplement 5*).

## Hydrophobic residues of the EMB domain are involved in membrane tubulation

The aforementioned structural analysis indicated that the S-OPA1 EMB domain is presumably responsible for membrane binding and deformation. According to sequence analysis, such an EMB domain corresponded to the residues from 738 to 853 in S-OPA1 (*Figure 3—figure supplement 1*). Among those residues, we identified a region (794-ELEKMLK-800) with interval hydrophobic and hydrophilic residue arrangements that might be involved in direct interaction with the membrane. Therefore, several mutations were produced on this region to investigate its role in membrane tubulation.

We constructed five S-OPA1 mutants by mutating regions 794–800 to all alanine residues (794-800A) or by inducing double- and triple-point mutations (E794AE796A, K797AK800A, L795EM798EL799E, and L795AM798AL799A). Compared with the WT S-OPA1, all five mutants

exhibited moderate suppression of membrane binding, suggesting that the residues of membrane binding were at least partially involved (*Figure 4A*). However, more considerable effects of these mutants were observed regarding their liposome-induced GTPase activity (*Figure 4B*; *Supplementary file 1*; *Source data 1*) and membrane tubulation activity (*Figure 5C*). The mutations 794-800A, L795EM798EL799E, and L795AM798AL799A completely abolished tubulation activity and liposome-induced GTP hydrolysis activity, whereas these activities were only partially reduced in

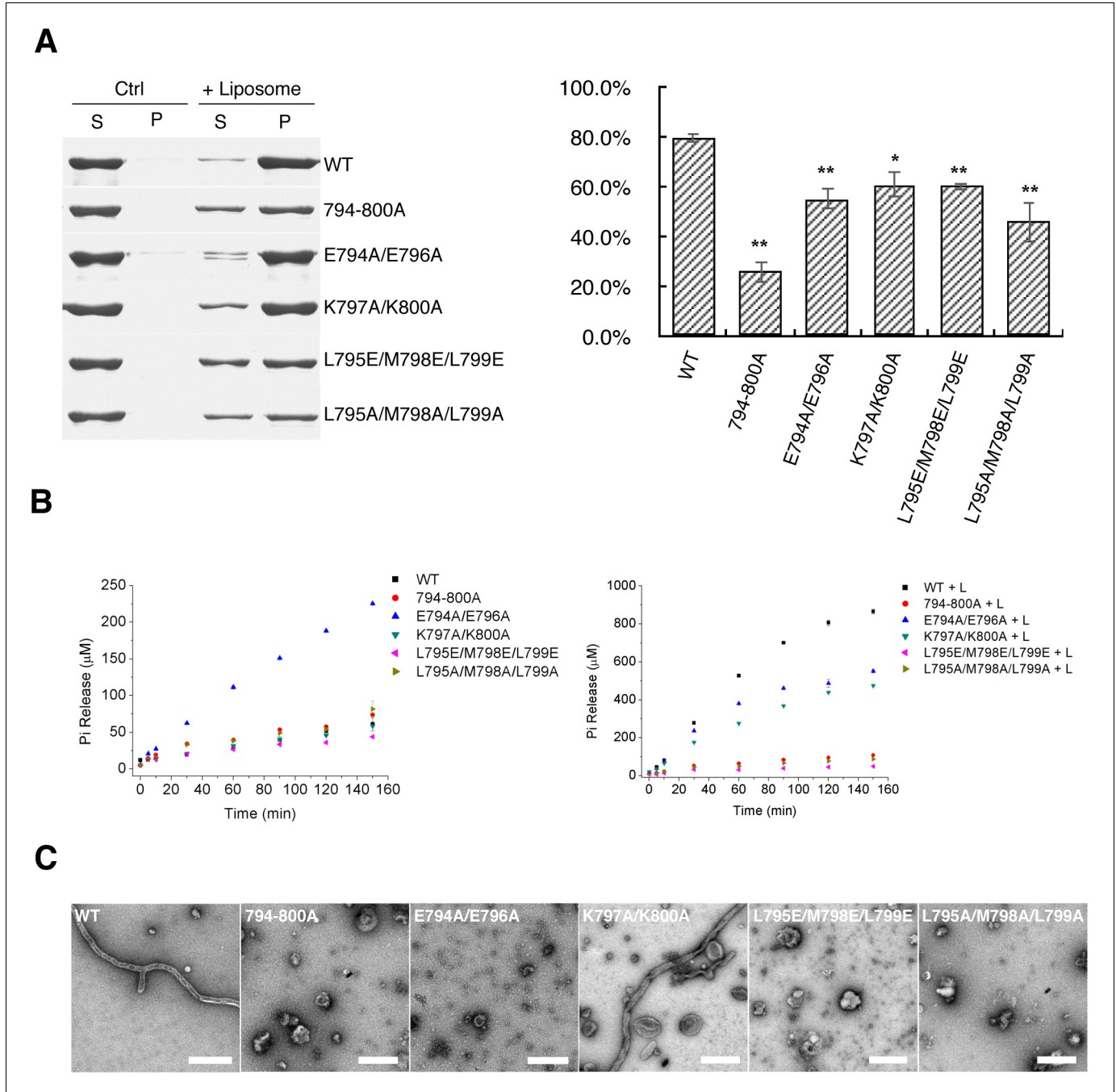

**Figure 4.** Mutants in EMB domain of S-OPA1 and their impact on tubulation activity. (**A**) Sedimentation of wild type S-OPA1 and its EMB domain mutants with or without cardiolipin containing liposomes (n = 3). S, supernatant; P, pellet; *, p<0.01; **, p<0.001. (**B**) Basal and liposome binding induced GTPase activity of wild type S-OPA1 and its EMB domain mutants. L, liposome. The total free phosphate was measured at each time point and data presented come from 3 independent experiments. See also *Source data 1*. (**C**) Tubulation activity of wild type S-OPA1 and its EMB domain mutants examined by negative stain electron microscopy. Scale bar, 500 nm.

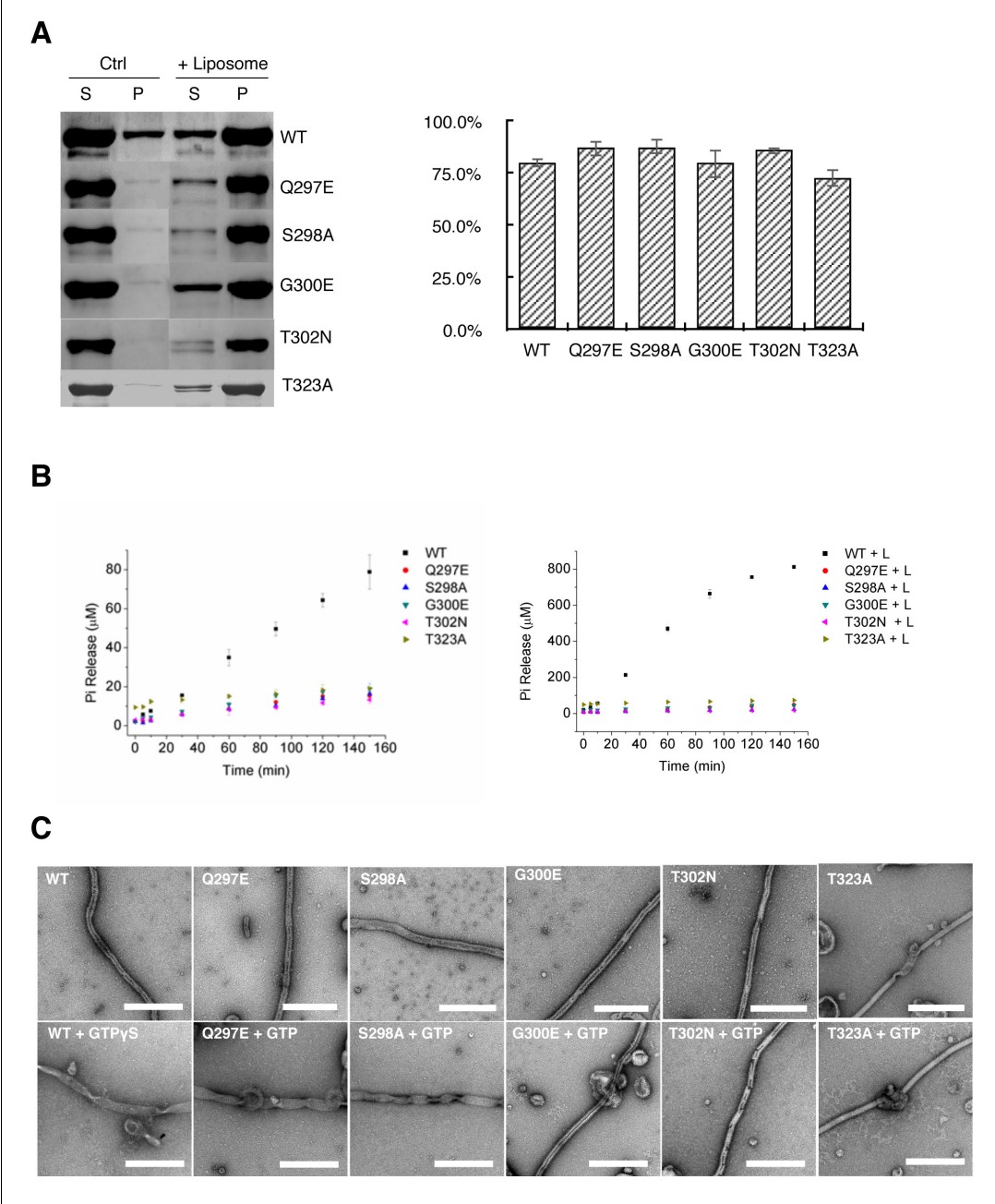

**Figure 5.** Tubulation activity of S-OPA1 is independent with its GTPase activity but depends on GTP binding. (**A**) Sedimentation of wild type S-OPA1 and its G domain mutants with or without cardiolipin containing liposomes (n = 3). S, supernatant; P, pellet; *, p<0.01; **, p<0.001. (**B**) Basel (left panel) and liposome binding induced (right panel) GTPase activity of wild-type S-OPA1 and its G domain mutants. L, liposome. The total free phosphate was measured at each time point and data presented result from 3 independent experiments. See also *Source data 1*. (**C**) Tubulation activity of wild type S-OPA1 and G domain mutants without (top) or with 1 mM GTP (bottom) examined by negative stain electron microscopy. Scale bar, 500 nm.
The online version of this article includes the following figure supplement(s) for figure 5:

**Figure supplement 1.** Cryo-EM analysis of S-OPA1 coated tubes after incubation with different nucleotides.
**Figure supplement 2.** Diameter distributions of S-OPA1 G domain mutants coated tubes with GTP or without (Apo).

E794AE796A and K797AK800A. These results indicate that the hydrophobic residues in region 794–800 of the EMB domain could be a crucial factor involved in S-OPA1–induced membrane tubulation. Furthermore, the reduced or abolished membrane tubulation activity then prevented the formation

of regular helical assembly, which consequently reduced membrane-stimulated GTP hydrolysis activity.

## S-OPA1 membrane tubulation activity is independent of its GTPase activity

Our observation that S-OPA1 can induce tubulation of liposomes without the addition of nucleotide suggests that the tubulation activity of S-OPA1 is independent of its GTP hydrolysis activity. To validate this assumption, we performed mutagenesis. The high sequence conservation of G domains among dynamin proteins enabled us to identify the locations (a.a. 297–302 and 319–327) of the key catalytic residues of S-OPA1 (*Figure 3—figure supplement 1*) at its P loop and switch I (*Schmid and Frolov, 2011*). All the mutants (Q297E, S298A, G300E, T302N, and T323A) maintained their WT abilities for liposome binding but had impaired liposome-stimulated GTPase activity (*Figure 5A and B*; *Supplementary file 1*; *Source data 1*). However, notably, their tubulation activity did not change considerably (*Figure 4C*). These results confirm that S-OPA1 can deform membranes without requiring GTP hydrolysis and further suggest that GTP hydrolysis of S-OPA1 most likely occurs after liposomal tubulation.

## Nucleotide binding leads to reduced membrane curvature

The GTPase activity of S-OPA1 is indispensable for promoting mitochondrial inner membrane fusion (*Ban et al., 2010*). However, the tubulation activity of S-OPA1 is independent of its GTPase activity. Considering that substantial conformational changes after nucleotide binding have been observed for Dyn1 and Dnm1 (*Chappie et al., 2011*; *Fröhlich et al., 2013*; *Mears et al., 2011*), investigating how nucleotide binding and hydrolysis affect the assembly of S-OPA1 might yield useful results.

We incubated excess GTP or its non-hydrolyzable or slowly hydrolyzable analogs (GMPPCP, GMPPNP, and GTPγS) with the WT- and truncated S-OPA1–coated tubes for 30 min before examination using cryo-EM (*Figure 5—figure supplement 1A*). Contrary to our expectations, the diameter of all tubes increased from the original diameter of approximately 53 nm to between 70 and 80 nm (*Figure 5—figure supplement 1B*). This contradicts the constriction phenomena observed in Dyn1 and Dnm1 but is similar to the prediction of S-Mgm1 (*Chappie et al., 2011*; *Faelber et al., 2019*; *Fröhlich et al., 2013*; *Mears et al., 2011*). In addition to expanding the tubes, the addition of nucleotides reduced their homogeneity (*Figure 5—figure supplement 1B*).

To further confirm that the S-OPA1 tube expansion was caused by nucleotide binding, we used the aforementioned mutants with defective GTPase activity (Q297E, S298A, G300E, T302N, and T323A) to investigate their tubulation behaviors after incubation with GTP. Studies have indicated that the mutants Q297E and S298A maintain their GTP binding activity (*Chappie et al., 2010*) whereas G300E, T302N, and T323A lose theirs (*Ban et al., 2010*; *Marks et al., 2001*; *Song et al., 2004*). Although all the mutants can bind and induce tubulation of liposome, the addition of GTP induced different results (*Figure 5C*; *Figure 5—figure supplement 2*). After we added GTP, the tubes coated with the mutants Q297E and S298A exhibited obvious changes, including increased diameter and heterogeneity, which is similar to those observed in the WT. By contrast, the tubes coated with the mutants G300E, T302N, and T323A did not respond remarkably to the addition of GTP. The diameter distribution of the T323A-coated tube exhibited a small shift after incubation with GTP; we speculate that this mutant might have retained a very weak binding affinity for GTP. These results further prove that nucleotide binding induces the expansion of S-OPA1 coated tubes.

## Helical assembly of S-OPA1 after nucleotide binding

The S-OPA1 coated liposomal tubes became unstable after incubation with GTP, most likely as a result of unsynchronized GTP hydrolysis. Thus, we selected the most stable and homogenous GTPγS binding state for structural study. The helical reconstruction technique failed because of the variable diameters. We therefore utilized cryo-ET and SVA to analyze the assembly of GTPγS-bound S-OPA1 on the membrane (*Figure 6—figure supplement 1*).

Similar to the nucleotide-free state, the 23 Å–resolution cryo-EM map of S-OPA1–coated tube bound with GTPγS also revealed four density layers: an outer head region, middle stalk region, inner leg region, and innermost membrane region (*Figure 6A*). However, after GTPγS binding, considerable gaps appeared between stalks in neighboring helical rungs as their interval expanded to

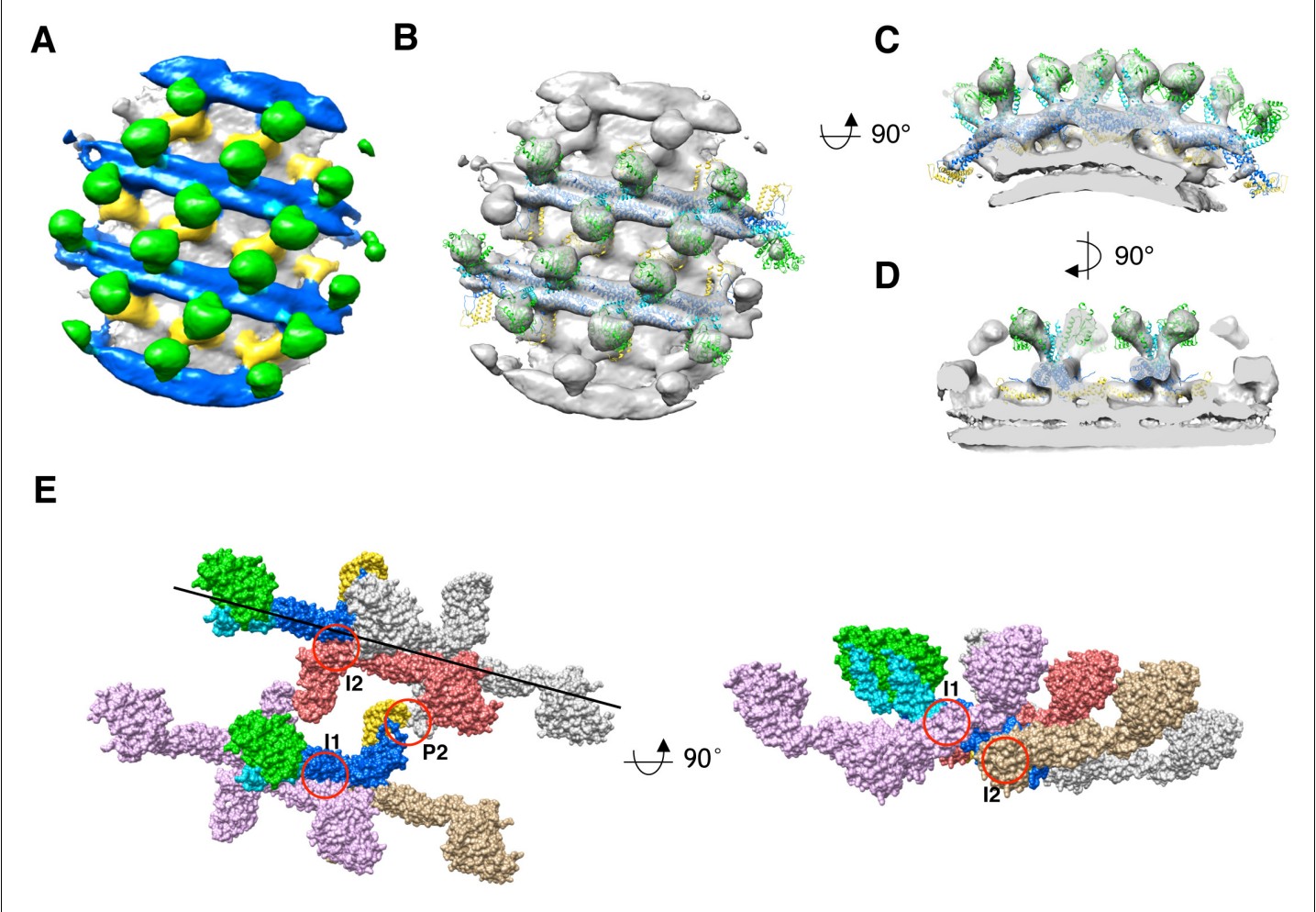

**Figure 6.** Sub-tomogram averaging of S-OPA1 coated tube at GTPγS binding state. (A) Side view of cryo-EM map of S-OPA1 coated tube after adding GTPγS. The map is subdivided into three layers and colored with the same scheme in *Figure 2*. (B) Docking of S-Mgm1 crystal structure into the map. Domains are colored as the same scheme in *Figure 3*. (C) Cross section view of the map that is horizontally rotated 90˚ from (B). (D) Cross section view of the map that is vertically rotated 90˚ from (C). (E) Structural model of S-OPA1 assembly on membrane at GTPγS binding state. The three interfaces (I1, I2 and P2) for the stability of S-OPA1 assembly are indicated with red circles. The orientation of the helical rung is indicated with the black line. The online version of this article includes the following figure supplement(s) for figure 6:

**Figure supplement 1.** Cryo-electron tomographic image processing of S-OPA1 coated tubes after adding GTPγS.

approximately 55 Å, which led to reduced compactness of the S-OPA1 assembly and a loosened helical lattice. In addition, the inner leg regions were observed to be located between helical rungs.

Subsequently, we docked the previously generated nucleotide-free state S-OPA1 model into this cryo-EM density (*Figure 6B-D*). The model fit well into the map, except for the need for some adjustment at the angle between G domain/BSE and stalk. The potential dimerization interfaces of G domains were still noted between neighboring rungs (*Figure 3—figure supplement 5*), however, the distance between the G domains was still too great to form a G dimer (*Figure 3—figure supplement 3*). This indicated a further conformational change for the subsequent GTP hydrolysis. The S-Mgm1 paddle domain fit well into the inner leg region, which identifies and localizes the EMB domain of S-OPA1 in the GTPγS binding state. Notably, the EMB/paddle domains of neighboring helical rungs seemed to form dimers between the stalk gaps.

By investigating the packing array of S-OPA1, we observed that the asymmetric unit of S-OPA1 nucleotide–coated tube in the GTPγS binding state also contained an S-OPA1 dimer (*Figure 6B and E*). This dimer had a similar interaction interface (interface-2) to that of the nucleotide-free state (*Figure 3E*). Here, interface-1 obviously mediated the interactions between S-OPA1 dimers within

the same helical rung (*Figure 6E*), whereas interface-3 in the nucleotide-free state was broken and disappeared. Notably, the interactions between different helical rungs in the GTPγS binding state were mediated through a new interface (interface P2) that formed between the EMB domains (*Figure 6E*). The assembly pattern of S-OPA1 in the GTPγS binding state was highly similar to that of S-Mgm1 in both apo and the GTPγS binding states, sharing the same position as interface-1 and interface-2 (*Figure 3—figure supplement 4*).

## Conformational change and helical assembly rearrangement after nucleotide binding

Subsequently, we further analyzed the conformational changes of S-OPA1 upon nucleotide binding by comparing the models generated from the nucleotide-free and GTPγS binding states. The superposition of two monomers indicated a slight (~4°) G domain/BSE swing after GTPγS binding (*Figure 7A*). In the GTPγS binding state, the S-OPA1 monomer tended to adopt a slightly open conformation compared with that adopted in the nucleotide-free state. This conformational change was not observed in S-Mgm1 (*Figure 7—figure supplement 1A*). After GTPγS binding, the angle between two stalks within the building block of the S-OPA1 dimer decreased by approximately 10°, and the center-to-center distance of the two monomers also decreased, leading to a more compact dimer conformation (*Figure 7B*; *Figure 7—figure supplement 1B*). Moreover, the S-OPA1 tetramer that formed through interface-1 and interface-2 exhibited more pronounced changes after GTPγS binding. After superimposition along the helical axis, we observed a clockwise rotation (~25°) of the S-OPA1 tetramer after GTPγS binding (*Figure 7C*; *Figure 7—figure supplement 1C*) and an approximately 80 Å decrease in tetramer extension (*Figure 7D*).

After GTPγS binding, interface-1 and interface-2 in S-OPA1 assembly generally maintained their original positions, with only a slight change in location of interface-2. However, interface-3 and interface P1 in the nucleotide-free state were broken, and a new inter-rung interface P2 formed between the EMB domains. As a result, for the left-handed model of S-OPA1 dimer assembly, GTPγS binding broke the original helical rungs and formed new rungs through combination of the two nearest pieces (*Figure 7E*). By contrast, for the right-handed model, the helical rungs remained, but the handedness of the assembly changed from right to left after GTPγS binding (*Figure 7E*). Further high-resolution structural studies are required to discriminate these two possibilities.

Overall, although the S-OPA1 assembly in the nucleotide-free state appeared more compact, its building unit adopted a more relaxed conformation with each S-OPA1 molecule in a closed state. After GTPγS binding, S-OPA1 monomers adopted an open conformation, which constrained the spread conformation of building units (dimers and tetramers) and finally resulted in a loosened helical rung and an expansion of the helical tube.

## Discussion

Membrane fission and fusion are critical for organelle communication, matter exchange, and cargo transportation in eukaryotic cells. Compared with vesicles and other organelle, mitochondria have even more complicated fission and fusion mechanisms because of their double-membrane structure. An increase in the amount of structural information regarding Mfn1/2 and Drp1/Dnm1 has spurred extensive research on the molecular mechanisms of mitochondrial outer membrane fusion and fission (*Cao et al., 2017*; *Fröhlich et al., 2013*; *Kalia et al., 2018*; *Mears et al., 2011*; *Qi et al., 2016*; *Yan et al., 2018*). However, the dearth of structural information for OPA1 has limited further interpretation of the mitochondrial inner membrane fusion mechanism. Compared with the general structure of Mfn1/2, that of OPA1 tends to be more similar to that of the membrane fission protein Dyn1. How a fission dynamin-like protein triggers membrane fusion remains unclear. Unlike other dynamins, OPA1 has multiple isoforms and must be processed from a long membrane–anchored form (L-OPA1) to a short soluble form (S-OPA1) for efficient mitochondrial inner membrane fusion. This suggests that the dynamics of the mitochondrial inner membrane are strictly regulated. Structural studies of *C. thermophilum* S-Mgm1 (CtMgm1) (*Faelber et al., 2019*) and *S. cerevisiae* S-Mgm1 (ScMgm1) (*Yan et al., 2020*), which are homologs of S-OPA1, have provided initial insights into the functional mechanism of S-Mgm1 during mitochondrial inner membrane fusion.

In the present study, we investigated the interactions between S-OPA1 and liposomes that have the phospholipid composition of the mitochondrial inner membrane. Similar to other dynamin

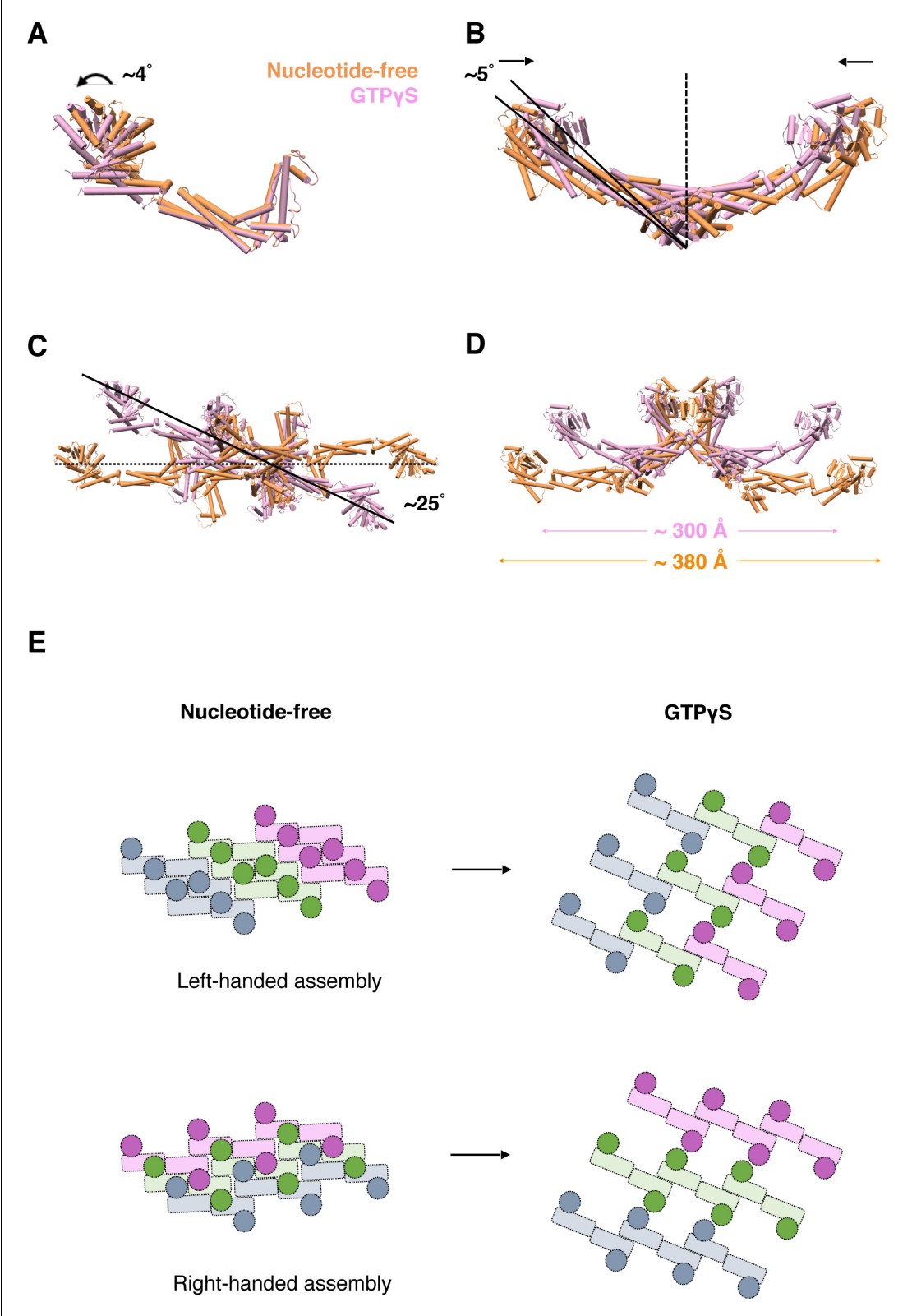

**Figure 7.** Conformational change of S-OPA1 and its assembly after GTPγS binding. (**A**) Conformational change of S-OPA1 monomer after GTPγS binding. Black arrows indicate the direction of the conformational change. (**B**) Conformational change of S-OPA1 dimer after GTPγS binding. (**C**) Conformational change of S-OPA1 tetramer after GTPγS binding. Angle change of the helical rung orientation is indicated. (**D**) Change of distance between two distal G domains within a single S-OPA1 tetramer. (**E**) Schemes of the helical assemblies of S-OPA1 on membrane at nucleotide-free and

*Figure 7 continued on next page*

*Figure 7 continued*

GTPγS binding states. The G domain is shown as a filled circle and the stalk region is shown as a filled rectangle. For nucleotide-free state, the S-OPA1 molecules locating in the same rung are colored same.

The online version of this article includes the following figure supplement(s) for figure 7:

**Figure supplement 1.** Comparison between S-OPA1 and *C. thermophilum* S-Mgm1 models in two different nucleotide binding states.

proteins, S-OPA1 can bind to membranes and induce membrane tubulation by forming a helical array. We revealed that such tubulation processes are independent of GTP binding and hydrolysis activity; however, S-OPA1 can exhibit considerably enhanced GTPase activity after binding to the membrane. We used the cryo-EM approach to solve the structures of S-OPA1 coated on membranes in both nucleotide-free and GTPγS binding states. Docking of an S-Mgm1 crystal structure into cryo-EM maps revealed that S-OPA1 has a classic dynamin-like structure that contains a G/BSE domain, a stalk domain, and an EMB domain. The EMB domain has a similar size and shape as the paddle domain of CtMgm1 and the LIS domain of ScMgm1.

Structural analysis revealed four interaction interfaces that contribute to S-OPA1 assembly on the membrane in the nucleotide-free state. Three interfaces (I1, I2, and I3) are involved in stalk packing. The fourth interface (P1) from the EMB domain interaction constitutes another factor influencing the stability of S-OPA1 assembly. In contrast to the tube formed by CtMgm1, which was left-handed with four starts, the tube formed by S-OPA1 in the absence of nucleotides was also left-handed but with six starts. In addition, CtMgm1 only used interface-1 and interface-2 to form the helical filament in both the nucleotide-free and binding states, whereas the helical assembly of S-OPA1 used additional interface-3 and interface P1 in the nucleotide-free state. We further identified a key region (794–800) of the EMB domain and observed that the hydrophobic residues of this region are instrumental to membrane tubulation activity. The observed critical role of the stalk and EMB domains for S-OPA1 assembly and membrane interaction is consistent with the findings of other studies that have highlighted the relationship between diseases and mutations in the stalk and EMB regions (*Supplementary file 2*). The G domain dimerization interface, which is crucial for GTP hydrolysis activity, was not formed in the helical array of S-OPA1 in the nucleotide-free state, suggesting that GTP hydrolysis is a late-stage event that occurs after membrane remodeling.

We observed substantial changes in S-OPA1 assembly after GTPγS binding, including motion between the G domain and the stalk, global rotation of the stalk region, and rearrangement of the S-OPA1 helical lattice. Here we defined the conformation of S-OPA1 in its nucleotide-free state as closed conformation and that in its GTPγS-bound state as open conformation. In contrast to conventional dynamin proteins that induce more crowded packing and a constricted tube with a smaller diameter and higher curvature (ready for fission) after nucleotide binding (*Chappie et al., 2011*; *Mears et al., 2011*; *Sundborger et al., 2014*), the left-handed helical assembly and the conformational change in S-OPA1 from closed to open after GTPγS binding resulted in an expanded tube with an increased diameter and reduced membrane curvature, which is in agreement with the proposed action of CtMgm1 (*Faelber et al., 2019*). The helical assemblies of S-OPA1 in the nucleotide-free and the GTPγS binding states were remarkably different. This phenomenon has not been reported in related studies of other dynamin proteins. After GTPγS binding, the dimer building blocks likely adopt a rotation of 25° and a long-range movement, which breaks interface-3 and interface P1 through the formation of a new interface, P2, between the EMB domains. This large movement might be triggered by the energy released from nucleotide binding. However, we could not exclude another possibility, namely that the dissociation and reassembly of S-OPA1 monomers occur after GTPγS binding.

A mechanochemical mechanism of the protein-induced membrane remodeling adopted by Dyn1 has been hypothesized (*Chappie et al., 2011*). The protein uses the energy generated from binding and hydrolysis of GTP to realize a conformational change. Such a change deforms the membrane and successfully transform the chemical energy into mechanical changes. Because S-OPA1 exhibited a similar G domain conformational change after nucleotide binding, it would probably employ the same mechanism in membrane remodeling. Although the resolution achieved by cryo-ET and subtomogram averaging was not high, our observation suggests that the distance between the S-OPA1 G domains between the two neighboring S-OPA1 dimers was shorter after GTPγS binding; however,

this distance was still too great for G domain dimerization to occur. Thus, the determined structure of the GTPγS-bound S-OPA1 is not compatible with GTP hydrolysis induced by G domain dimerization. This might be attributable to a subtle difference between GTPγS and GTP, which constitutes a barrier to future structural reorganization of S-OPA1 helical arrays to induce the dimerization of G domains. Thus, we propose a further motion of the G domain upon GTP binding, which enables the formation of a G domain dimerization interface that is suitable for GTP hydrolysis.

Previous studies have revealed that S-OPA1 alone cannot trigger the fusion of the mitochondrial inner membrane but rather must collaborate with L-OPA1 (*Ban et al., 2017*; *Del Dotto et al., 2017*; *Song et al., 2007*). During mitochondrial inner membrane fusion, L-OPA1 performs a critical role, and the addition of S-OPA1 significantly increases fusion efficiency (*Ban et al., 2017*). In addition, the GTPase activity of S-OPA1 is indispensable for its fusion-related function (*Ban et al., 2017*). Therefore, S-OPA1 presumably assists L-OPA1 with its GTPase activity during mitochondrial inner membrane fusion. Thus, we speculate that S-OPA1 facilitates membrane fusion through oligomerization with L-OPA1, thus supporting GTP hydrolysis and conformational change of L-OPA1. Research on Mgm1 has revealed a similar situation in which the GTPase activity of L-Mgm1 is inhibited because of the restriction of G domain movement by the transmembrane helix (*DeVay et al., 2009*). S-Mgm1 may then bind to L-Mgm1 to facilitate the formation of G dimers with accelerated GTPase activity.

Although our structural and functional analyses demonstrated the assembly of S-OPA1 on membranes in nucleotide-free and GTPγS binding states and provided models of how the rearrangement of this assembly affects membrane remodeling, the precise molecular mechanism of S-OPA1–induced membrane fusion and the cooperation between L-OPA1 and S-OPA1 require further study. High-resolution structural information regarding S-OPA1 and L-OPA1 in different GTP hydrolysis states could help to clarify the mechanisms of mitochondrial inner membrane fusion.

# Materials and methods

## Key resources table

| Reagent type (species) or resource | Designation | Source or reference | Identifiers | Additional information |
|---|---|---|---|---|
| Chemical compound, drug | Palmitoyl-2-oleoyl-sn-glycero-3-phosphocholine (POPC) | Avanti Polar Lipids | Cat#: 850375 | |
| Chemical compound, drug | 1-palmitoyl-2-oleoyl-sn-glycero-3-phosphoethanolamine (POPE) | Avanti Polar Lipids | Cat#: 850725 | |
| Chemical compound, drug | L-α-lysophos phatidylinositol (PI) | Avanti Polar Lipids | Cat#: 840042 | |
| Chemical compound, drug | 1′,3′-bis[1,2-dioleoyl-sn-glycero-3-phospho]-sn-glycerol (cardiolipin) | Avanti Polar Lipids | Cat#: 840012 | |
| Chemical compound, drug | Guanosine 5′-triphosphate sodium salt hydrate (GTP) | Sigma-Aldrich | Cat#: G8877 | |
| Chemical compound, drug | Guanosine 5′-diphosphate sodium salt (GDP) | Sigma-Aldrich | Cat#: G7127 | |
| Chemical compound, drug | Guanosine 5′-[γ-thio] triphosphate tetralithium salt (GTPgammaS) | Sigma-Aldrich | Cat#: G8634 | |

*Continued on next page*

*Continued*

| Reagent type (species) or resource | Designation | Source or reference | Identifiers | Additional information |
|---|---|---|---|---|
| Chemical compound, drug | Guanosine 5'-[β,γ-imido] triphosphate trisodium salt hydrate (GMPPNP) | Sigma-Aldrich | Cat#: G0635 | |
| Chemical compound, drug | β,γ-Methyleneguanosine 5'-triphosphate sodium salt (GMPPCP) | Sigma-Aldrich | Cat#: M3509 | |
| Chemical compound, drug | Bis (sulfosuccinimidyl) suberate (BS3) | Thermo Fisher Scientific | Cat#: 21580 | |
| Software, algorithm | IMOD | IMOD(https://bio3d.colorado.edu/imod/) | RRID:WB-STRAIN:WBStrain00027352 | Version 4.9 |
| Software, algorithm | RELION | RELION(https://www3.mrc-lmb.cam.ac.uk/relion/index.php?title=Main_Page) | RRID:SCR_016274 | Version 1.4 |
| Software, algorithm | SPIDER | SPIDER(https://spider.wadsworth.org/spider_doc/spider/docs/spider.html) | | Version 24.08 |
| Software, algorithm | EMAN | EMAN(https://blake.bcm.edu/emanwiki/EMAN2) | RRID:SCR_016867 | Version 2.3.1 |
| Software, algorithm | SerialEM | Serial(http://bio3d.colorado.edu/SerialEM/) | RRID:SCR_017293 | Version 3.7 |
| Software, algorithm | Chimera | Chimera(http://plato.cgl.ucsf.edu/chimera/) | RRID:SCR_004097 | Version 1.14 |
| Software, algorithm | IHRSR | IHRSR(https://www.ncbi.nlm.nih.gov/pmc/articles/PMC3245864/) | | Edward H. Egelman, University of Virginia, Virginia, USA |
| Software, algorithm | AuTom | AuTom(https://www.sciencedirect.com/science/article/pii/S1047847717301284) | | Renmin Han, King Abdullah University of Science and Technology, Thuwal, Saudi Arabia |
| Software, algorithm | GCTF | GCTF(https://www.mrc-lmb.cam.ac.uk/kzhang/Gctf/) | RRID:SCR_016500 | Version 1.06 |
| Software, algorithm | MotionCor2 | MotionCor2(https://emcore.ucsf.edu/cryoem-software) | RRID:SCR_016499 | |
| Software, algorithm | CTFFIND | CTFFIND( http://grigorifflab.janelia.org/ctffind4) | RRID:SCR_016732 | |

## Protein expression and purification

cDNA corresponding to the short S1 isoform of OPA1 was sub-cloned into the pET32M-3C expression vector (from Wei Feng's Lab, IBP, CAS) with a N-terminal Trx tag and a followed His6 tag. A PreScission protease cleavage site exists between the His tag and the coding sequence. Mutants were constructed in pET32M-3C/OPA1-S1 via PCR. All proteins were expressed in Transetta (DE3) bacteria cells (Transgene) and purified under the following procedure. Cultures were grown at 37℃ until OD at 600 nm reached 0.8. Protein expression was induced with 0.2 mM IPTG for 18 hr at 16℃. The cells were collected by centrifugation. Bacteria pellets were resuspended in lysis buffer containing 20 mM Tris-HCl (pH 8.0), 150 mM NaCl and protease inhibitors cocktail (Roche) and disrupted with ultra-sonication. Lysates were incubated with Ni-NTA beads (Roche). After washing with the buffer containing 20 mM Tris-HCl (pH 8.0), 150 mM NaCl, 1 mM DTT, and 10 mM imidazole, protein was cleaved by prescission protease at 4℃ overnight. After cleavage, protein was eluted with 20 mM Tris-HCl (pH 8.0), 150 mM NaCl, 1 mM DTT, 20 mM imidazole. The eluted protein fraction was

further purified by gel filtration chromatography using a Superdex 200 10/300 GL column (GE Healthcare) in the buffer of 20 mM Tris-HCl (pH 8.0), 150 mM NaCl, and 1 mM DTT. The elution volume of the column was pre-calibrated using standard protein molecular weight markers. Purified proteins were frozen in liquid nitrogen and stored at −80°C.

## Preparation of S-OPA1 coated tubes

The lipids (Avanti Polar Lipids) were mixed in the following ratio: 45% palmitoyl-2-oleoyl-sn-glycero-3-phosphocholine (POPC), 22% 1-palmitoyl-2-oleoyl-sn-glycero-3-phosphoethanolamine (POPE), 8% L-α-lysophosphatidylinositol (PI), and 25% 1′,3′-bis[1,2-dioleoyl-sn-glycero-3-phospho]-sn-glycerol (cardiolipin). The indicated ratios of lipids were mixed in a chloroform solution, evaporated for 4 hr in a vacuum desiccator and rehydrated in the buffer of 20 mM Tris–HCl (pH 8.0), 1 mM EGTA, and 1 mM MgCl$_2$ to a final concentration of 4 mg/ml. The resulting multi-lamellar liposomes were put through five freeze/thaw cycles to make unilamellar liposomes. S-OPA1 protein was then mixed with unilamellar liposomes 1:1 (m:m) at a final concentration of 1 mg/ml in the buffer containing 20 mM HEPES (pH 8.0), 1 mM EGTA, and 1 mM MgCl$_2$. The mixture was incubated at 16°C for 30 min before preparing cryo-EM samples. For tubes incubate with GTP and GTP non-hydrolyzed analogous, 10 mM nucleotide was then added with a final concentration of 1 mM, and the mixture was incubated for another 30 min.

For negative staining EM, 5 μl of protein-lipid tubes was applied to glow-discharged continuous carbon films and stained with uranyl acetate (2% w/v) for 1 min. Samples were visualized using a Tecnai Spirit electron microscope (ThermoFisher Scientific) operating at 120 kV and image were recorded with an Eagle camera.

Cryo-EM grids were prepared with Vitrobot Mark IV (ThermoFisher Scientific) under 100% humidity. 3 μl of protein-lipid tubes was applied to glow-discharged Quantifoil R2/1 holy carbon grids, blotted, and plunged into liquid ethane. For grids using for tomography data collection, homemade protein A coated colloidal gold was added as a fiducial marker.

## Cryo-electron microscopy

Images for helical reconstruction were recorded on a cryo-electron microscope Titan Krios (ThermoFisher Scientific) operating at 300kV using SerialEM software (*Mastronarde, 2005*). A Falcon-IIIEC camera (ThermoFisher Scientific) was used at a calibrated pixel size of 1.42 Å. A combined total dose of 50 e/Å$^2$ was applied with each exposure. Images were collected at 2–4 μm underfocus.

Tilt series data were collected on a cryo-electron microscope Titan Krios G2 (ThermoFisher Scientific) using SerialEM software (*Mastronarde, 2005*), with a K2 direct electron detector (Gatan) operating in counting mode. Tilt series data were typically collected from ±45° with 3° tilt increments at 3–5 μm underfocus. A combined dose of about 90 e/Å$^2$ was applied over the entire tilt series.

## Helical reconstruction

In total, 2112 movie stacks were collected. Motion correction and defocus estimation for all these micrographs were performed using MotionCorr2 (*Zheng et al., 2017*) and GCTF (*Zhang, 2016*) respectively. Micrographs with ice contamination, poor Thon rings, too large defocus values (greater than 3 μm) were excluded before tube boxing. Good micrographs were then multiplied by their theoretical contrast transfer function (CTF) for initial correction of CTF. 511 S-OPA1 tubes were boxed using e2helixboxer.py in the package of EMAN2 (*Tang et al., 2007*) with a 480 px box width. An initial segment was stack generated from all these tubes with an overlap of 90% and an initial 3D model was generated by back projection method using these segments and assigning random azimuthal angles to them. The initial 3D model was then interpolated into different ones with various diameters, and diameter classification was performed through supervised 2D classification where models were generated by projecting the 3D models with various diameters. Then for each diameter class, the diffraction pattern for each tube was calculated and further classified. A main class of tubes at ~53 nm diameter was sorted corresponding to its diameter and diffraction pattern, and contains 6644 segments. The segment stack of the selected class was then regenerated with the box size of 480 px and the box overlap of ~94%. Initial helical parameters were calculated by indexing the layer lines in the power spectrum of the boxed tubes. An initial helical rise of 27.0 Å and twist of −20.87° were obtained and used for helical reconstruction through a real space helical reconstruction

algorithm IHRSR (*Egelman, 2000*; *Egelman, 2007*). The helical parameters finally converged to 25.87 Å for the helical rise, and −20.86° for the helical twist. Then summed $CTF^2$ was divided for the final CTF correction of the map reconstructed by IHRSR. SPIDER (*Shaikh et al., 2008*) was used for negative B-factor sharpening. Resolution of the final map was estimated based on the gold standard Fourier shell correlation (FSC)0.143 criterion.

## Tomographic reconstruction and sub-volume averaging

Fiducial marker based tilt series alignment and gold erasure were performed using AuTom (*Han et al., 2017*). And the tomographic reconstructions were performed using IMOD (*Kremer et al., 1996*) with 2 times binning. No CTF correction was performed at this step. For tomographic reconstruction, the radial filter options were set at 0.35 cut off and 0.05 fall off. The sub volumes picked in IMOD were extracted by RELION 1.4 (*Scheres, 2012*) and CTF model of each particles was generated through RELION script that called CTFFIND4 (*Rohou and Grigorieff, 2015*). Then 3D classification was carried with CTF correction and the particles from selected classes were used for the final refinement. The initial model for 3D classification and refinement is the random averaging of all particles and low-pass filtered to 60 Å. Reported resolutions are based on the gold-standard Fourier shell correlation (FSC) 0.143 criterion.

## Sedimentation assay

Sedimentation assay was carried on as previous study (*Ban et al., 2010*). Protein was diluted to a concentration of 0.2 mg/ml in 20 mM Tris-HCl (pH 8.0), 300 mM NaCl, 1 mM $MgCl_2$, 1 mM EGTA, 1 mM DTT. Liposomes were prepared as described for the tubulation assay. The liposomes were directly added to the protein solution at a final concentration of 0.2 mg/ml and incubated at room temperature for 30 min. Samples were centrifuged at 250,000 g in a S140AT rotor (Hitachi) for 20 min at 4°C. The supernatant and pellet were analyzed by SDS–PAGE.

## GTPase activity assay

GTPase reactions were performed as previous studies (*Ban et al., 2010*) with 0.1 mg/ml protein and 0.1 mg/ml liposomes in 20 mM HEPES (pH 7.5), 1 mM EGTA, and 1 mM $MgCl_2$. GTP hydrolysis was quantified by monitoring the free phosphate concentration using a malachite green assay (*Leonard et al., 2005*). Reactions were initiated by the addition of GTP to 1 mM final concentration and incubated at 37°C. 20 µl mixtures were quenched with 5 µl of 0.5 M EDTA at regular time points. After the addition of 150 µl malachite green solution, the free phosphate concentration was monitored by the absorbance at 650 nm in a 96-well plate reader (EnSpire 2300).

Calculation of Km and Kcat is based on Lineweaver-Burk plot with the measurement at 6 different GTP concentration. The final GTP concentration was set to 0.4, 0.5, 0.6, 0.8, 1.0, and 1.4 mM individually. Reactions were initiated by the addition of 10 µl GTP solution to a 10 µl 0.1 mg/ml protein or protein liposome mixtures in 20 mM HEPES (pH 7.5), 1 mM EGTA, and 1 mM MgCl2. After 1 ~ 2 hr incubation, which was set based on the activity of different mutants, at 37°C, the reactions were quenched with 10 µl 0.1 M EDTA. The phosphate was quantified using malachite green assay. 150 µl malachite green solution was added and the absorbance at 650 nm is monitored in a 96-well plate reader (EnSpire 2300). All experiments were repeated at least three times.

## Size exclusion chromatography coupled with multi angle light scattering

The size of S-OPA1 in solution is determined by static multiangle light scattering (MALS) coupled with gel filtration. The size-exclusion chromatography column (PROTEIN KW-803, Shodex) is equilibrated with 20 mM Tris (pH 8.0), 150 mM NaCl, and 100 µl of 1 mg/ml purified S-OPA1 was applied. The detector DAWN HELEOS II (Wyatt) was used to measure the mass distribution. Data were analyzed using the provided ASTRA software.

## Chemical crosslink assay

S-OPA1 and Δ196–252 was diluted to 0.3 mg/ml in 50 mM HEPES (pH 7.4), 300 mM NaCl and 1 mM DTT. The amine-reactive crosslinker bis(sulfosuccinimidyl) suberate (BS3; Thermo Fisher Scientific) was added to a final concentration of 50 µM. After a 15 min incubation at room temperature, the

crosslinking reaction was quenched with 50 mM Tris (pH 8.0). Crosslinked products were analyzed using a 3–8% Tris–acetate PAGE.

## Acknowledgements

We would like to thank Prof. Edward H Egelman from University of Virginia for his generous help on image processing of helical reconstruction. We are grateful to Xue Wang (FS lab) for her initial trials of tubulation experiments. We would also like to thank Ping Shan and Ruigang Su (FS lab) for their assistances. This manuscript was edited by Wallace Academic Editing.

This work was supported by National Natural Science Foundation of China (31770794), grants from Chinese Academy of Sciences (XDB08030202) and the Ministry of Science and Technology of China (2017YFA0504700). All the EM works were performed at Center for Biological Imaging (CBI, http://cbi.ibp.ac.cn), Institute of Biophysics, Chinese Academy of Sciences.

## Additional information

### Funding

| Funder | Grant reference number | Author |
| --- | --- | --- |
| National Natural Science Foundation of China | 31770794 | Yan Zhang |
| Chinese Academy of Sciences | XDB08030202 | Fei Sun |
| Ministry of Science and Technology of the People's Republic of China | 2017YFA0504700 | Fei Sun |

The funders had no role in study design, data collection and interpretation, or the decision to submit the work for publication.

### Author contributions

Danyang Zhang, Software, Formal analysis, Visualization, Methodology; Yan Zhang, Software, Data processing, Paper writing, Formal analysis, Funding acquisition, Visualization; Jun Ma, Resources; Chunmei Zhu, Investigation; Tongxin Niu, Software; Wenbo Chen, Resources, Methodology; Xiaoyun Pang, Yujia Zhai, Validation, Methodology; Fei Sun, Conceptualization, Formal analysis, Supervision, Funding acquisition, Project administration

### Author ORCIDs

Danyang Zhang (iD) https://orcid.org/0000-0002-7279-9267
Yan Zhang (iD) https://orcid.org/0000-0001-5575-6882
Fei Sun (iD) https://orcid.org/0000-0002-0351-5144

### Decision letter and Author response

Decision letter https://doi.org/10.7554/eLife.50294.sa1
Author response https://doi.org/10.7554/eLife.50294.sa2

## Additional files

### Supplementary files

• Source data 1. Raw data of enzymatic assays of S-OPA1 and its mutants as well as detailed data processing.

• Supplementary file 1. Enzymatic Km and Kcat of S-OPA1 and its mutants. The raw data come from three independent experiments.

• Supplementary file 2. Disease related mutants of S-OPA1. Data was obtained from UniProt (https://www.uniprot.org).

• Transparent reporting form

## Data availability

Cryo-EM maps of S-OPA1 196-252 coated tubes have been deposited into Electron Microscopy Data Bank with the accession codes EMD-9901 for the helical reconstruction of nucleotide-free state, EMD-9903 for the tomographic reconstruction of nucleotide-free state and EMD-9902 for the tomographic reconstruction of GTPγS bound state, respectively. Sub-tomogram averaged cryo-EM map of wild type S-OPA1 coated tubes is also deposited with the accession code of EMD-0722. The raw data of GTPase assay in this study has been included as a supporting file.

The following datasets were generated:

| Author(s) | Year | Dataset title | Dataset URL | Database and Identifier |
|---|---|---|---|---|
| Zhang D, Zhang Y, Sun F | 2019 | Helical reconstruction of S-OPA1 at nucleotide-free state | http://www.ebi.ac.uk/pdbe/entry/emdb/EMD-9901 | Electron Microscopy Data Bank, EMD- 9901 |
| Zhang D, Zhang Y, Sun F | 2019 | S-OPA1 coated liposome tube at GTPgamaS bound state | http://www.ebi.ac.uk/pdbe/entry/emdb/EMD-9902 | Electron Microscopy Data Bank, EMD- 9902 |
| Zhang D, Zhang Y, Sun F | 2019 | S-OPA1 coated liposome tube at nucleotide-free state | http://www.ebi.ac.uk/pdbe/entry/emdb/EMD-9903 | Electron Microscopy Data Bank, EMD- 9903 |
| Zhang D, Zhang Y, Sun F | 2019 | Full length S-OPA1 coated liposome tube at nucleotide-free state | http://www.ebi.ac.uk/pdbe/entry/emdb/EMD-0722 | Electron Microscopy Data Bank, EMD- 0722 |

The following previously published datasets were used:

| Author(s) | Year | Dataset title | Dataset URL | Database and Identifier |
|---|---|---|---|---|
| Faelber K, Dietrich L, Noel JK, Wollweber F, Pfitzner AK, Muehleip A, Sanchez R, Kudryashev M, Chiaruttin N, Lilie H, Schlegel J, Rosenbaum E, Hessenberger M, Matthaeus C, Noe F, Roux A, vanderLaan M, Kuehlbrandt W, Daumke O | 2019 | Crystal structure of nucleotide-free Mgm1 | https://www.rcsb.org/structure/6QL4 | RCSB Protein Data Bank, 6QL4 |
| Faelber K, Dietrich L, Noel JK, Sanchez R, Kudryashev M, Kuehlbrandt W, Daumke O | 2019 | Structure of s-Mgm1 decorating the outer surface of tubulated lipid membranes | https://www.rcsb.org/structure/6RZT | RCSB Protein Data Bank, 6RZT |
| Faelber K, Dietrich L, Noel JK, Sanchez R, Kudryashev M, Kuelbrandt W, Daumke O | 2019 | Structure of s-Mgm1 decorating the outer surface of tubulated lipid membranes in the GTPgammaS bound state | https://www.rcsb.org/structure/6RZU | RCSB Protein Data Bank, 6RZU |

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
