## [Decision Letter]

**Acceptance summary:**

This study reveals two cryo-EM structures of truncated human OPA1 helical polymers on tubulated liposomes in both nucleotide-free and GTP-γ-S-bound forms. In combination with biochemical approaches, this study provides important information on our understanding of how OPA1 remodels the inner mitochondrial membrane.

**Decision letter after peer review:**

Thank you for submitting your article "Cryo-1 EM structures reveal interactions of S-OPA1 with membrane and changes upon nucleotide binding" for consideration by *eLife*. Your article has been reviewed by three peer reviewers, and the evaluation has been overseen by Hitoshi Nakatogawa as the Reviewing Editor and Olga Boudker as the Senior Editor.

The reviewers have discussed the reviews with one another, and the Reviewing Editor has drafted this decision to help you prepare a revised submission.

Summary:

In this manuscript, Zhang et al. report two cryo-EM structures of truncated human OPA1 helical polymers on tubulated liposomes in both nucleotide-free and GTP-γ-S-bound forms. They also investigated the liposome binding and tubulation ability of Opa1 by biochemical approaches. Major findings in this study are the assembly of Opa1 polymer, and its conformational change upon its GTP-γ-S binding. As properly cited by the authors of this manuscript, recently the crystal and cyro-EM structures of yeast Opa1 homolog Mgm1 were reported. Nonetheless, the reviewers found that the structure characterization presented in this paper, with considerable technical merit, still holds sufficient conceptual advance in a mammalian homolog, and provides important information on our understanding of how OPA1 remodels the inner mitochondrial membrane. However, the reviewers have also raised a number of critical issues which the authors should address to improve the integrity of the current manuscript and strengthen their conclusions.

Essential revisions:

As essential revisions, we request the authors to address the following issues.

1) Figure 2: handedness, fit of the nucleotide-free map. There is a mismatch of the statement of a "left-handed helical map" with the stalk filament fitted in a right-handed manner (Figure 7C). To avoid confusion, the authors should consistently state the handedness of their maps based on the assembly of the stalk filament.

Nevertheless, it is doubtful that the right-handed stalk assembly fitted into the nucleotide-free map, is correct, for the following reasons:

i) None of the interfaces observed in this filament has been described before for any dynamin family member.

ii) The map is only of low resolution and an unambiguous fitting of the domains is not straightforward.

iii) The new 'interfaces' have not been confirmed by any mutation.

iv) The suggested rotation of the OPA1 dimer upon nucleotide-binding would lead to steric clashes with neighboring molecules of the same filament and with molecules from the adjacent filament (Figure 7C).

Looking at the density map and the fitted models (EMD-9901, helical reconstruction), there appears to be a second option for fitting of the nucleotide-free map. Such an assembly would indeed constitute a left-handed stalk filament. It would also contain the same oligomerisation interfaces as in the GTP-γ-S bound OPA1 oligomer, and these interfaces would be similar to the previously described assembly interfaces 1 and 2 of dynamin (Ford et al., 2011, Faelber et al., 2011), MxA/B (Gao et al., 2011, Alvares, 2017), and Mgm1 (Faelber et al., 2019). Furthermore, such left-handed helical stalk assembly may explain why GTP-γ-S binding leads to expansion of the membrane tube and not constriction, unlike in dynamin (see the recent Mgm1 structure paper for discussion of helix handedness and the effect of the powerstroke). Finally, such assembly would explain the transition between the nucleotide-free and nucleotide-bound OPA1 assembly.

For both the nucleotide-free and the nucleotide-bound maps, the stalk has to be rotated for 180 degree around its longest axis that interface-1 and interface-2 can form in a similar manner compared to dynamin and Mgm1.

This should be illustrated in a detailed comparison in Supplementary Figure 8. For example, the OPA1 assembly mode contrasts with dynamin, MxA and Drp1, which use three assembly interfaces, but is similar to Mgm1, that requires only two interfaces.

Taken together, the current fitting of the nucleotide-free map is ambiguous and the resulting stalk assembly is unseen in the dynamin family and thus appears highly speculative. The authors should consider alternative fittings for both maps.

2) As now the structures of the yeast Opa1 homolog Mgm1 are available, the authors are suggested to use the Mgm1 model to fit the cryo-EM density and make corresponding analysis. Considering the substantial difference between the Mgm1 paddle domain and dynamin PH domain, as well as the special kink of the Mgm1 stalk domain, more accurate models could be obtained by this analysis. In this case, the authors may add also the Mgm1 sequence into the alignment in Supplementary Figure 5A.

3) As the structural difference between the nucleotide-free and GTP-γ-S bound states of Opa1 polymer is a major finding of this manuscript, the author should perform an ITC analysis to confirm and quantify the association between Opa1 and GTP-γ-S.

4) Supplementary Figure 5: GMB comparison with PH domain in dynamin. A sequence alignment of dynamin and OPA1 shows a very low amino acid identity in the region of the PH and GMB domain. A reliable alignment is not possible in this region. The authors speculate that amino acids 794-800 form an amphipathic helix which may bind to the membrane. However, the envisaged sequence is less than two turns in length, which would constitute only a small membrane anchoring point. Second, the proposed OPA1 amino acids are aligned to an amino acid stretch in the PH domain of dynamin, which forms two β-strands (compare sequence alignment to the structure of the PH domain, Ferguson et al., 1994, pdb 2dyn). In fact, the aligned β-strand region in the PH domain contains a known membrane binding loop. These observations do not support the conclusion that membrane binding in OPA1 is mediated by an amphipathic helix. The investigation of charged and hydrophobic residues in the GMB domain in order to detect the membrane binding region is sufficient to explain the experiments in Figure 5 without invoking an amphipathic helix.

5) Figures 1E and 4C: It should be clearly stated which nucleotide was used in which experiment and whether GTP-γ-S-induced structural changes were visible in negative stain EM. Representative negative-stain images should be provided for both conditions. The experiments with G interface mutants in Figure 4C make only sense in the presence of GTP-γ-S, since no structural changes can be expected for these mutants in the absence of nucleotide.

[Editors' note: further revisions were suggested prior to acceptance, as described below.]

Thank you for resubmitting your work entitled "Cryo-EM structures reveal interactions of S-OPA1 with membrane and changes upon nucleotide binding" for further consideration by *eLife*. Your revised article has been assessed by two reviewers who evaluated the original manuscript, and the evaluation has been overseen by the Reviewing Editor Hitoshi Nakatogawa and the Senior Editor Olga Boudker.

The reviewers found that the manuscript has been improved significantly but they raised some remaining issues that need to be addressed before final acceptance, as outlined below:

Essential revisions:

1) The assembly of the stalks in the nucleotide-free structure is still confusing. The new fittings, depiction of stalk interfaces and the schematic Figure 7E show that both the nucleotide-free and GTP-γ-S bound sOPA1 use interface-1 and 2 to form a filament. These interfaces are also used in Mgm1 and have been shown by mutagenesis to contribute to assembly. In contrast, the newly observed interface-3 is unique to the nucleotide-free form, is opposite of all previously known stalk interfaces, and due to missing mutagenesis, its significance for assembly remains unknown. To avoid confusion, the authors should use interface-1 and 2 to define the OPA1 filament also in the nucleotide-free form, e.g. in Figure 3A and E, Figure 3—figure supplement 2 and Figure 7E left. In Figure 7E left, not the blue and green dimers, but the blue striped and blue dimers should form the filament, as in Figure 7E right. In this case, I1 and I2 are present within a filament, and I3 is in between filaments. It would be helpful for the reader if one such turn is shown in a unique color in Figure 2A or Figure 3A (similar as it is now with the 'I3 filament') to indicate whether such filament is right- or left-handed. How do the G domains in this case connect? One might expect that the putative G domain interface is then in between different rungs.

2) Figure 4C: The new data on the GTP-bound sOPA1 mutants are nicely contributing to the manuscript. However, tube diameters have to be quantified for the mutants in the absence and presence of GTP (for example as in Figure 6—figure supplement 1 or as average diameter) to make a convincing claim. It also appears from Figure 1F that the representative tubes of the Δ196-252 variant are somehow smaller compared to sOPA1. Could the authors provide also some quantification of these?

3) Nomenclature GMB domain: The fitted paddle is anything but a globular domain (unlike the previously assumed PH domain). Maybe 'Extended Membrane Binding Domain' (EMB) instead?

4) Figure 1D legend should contain the used GTP concentration. It appears from Figure 4B that the stimulation of wt is about 10-fold, not 70-fold?

---

## [Author Response]

Essential revisions:As essential revisions, we request the authors to address the following issues.1) Figure 2: handedness, fit of the nucleotide-free map. There is a mismatch of the statement of a "left-handed helical map" with the stalk filament fitted in a right-handed manner (Figure 7C). To avoid confusion, the authors should consistently state the handedness of their maps based on the assembly of the stalk filament.Nevertheless, it is doubtful that the right-handed stalk assembly fitted into the nucleotide-free map, is correct, for the following reasons: […] The authors should consider alternative fittings for both maps.

We would like to thank the reviewers very much for this important critical points. Yes, the helical map of S-OPA1 coated tube is left-handed. The model in the previous Figure 7C is misleading. The apparent right-handed stalk region is just due to a visual effect from the compact packing of S-OPA1 on membrane in nucleotide-free state, in which the interaction between neighboring helical rung is a key factor for the stability of assembly.

As suggested by reviewers, we utilized the recent published crystal structure of S-Mgm1 (PDB code 6QL4) to re-perform structural fitting of both nucleotide-free and GTP-γ-S bound maps. With a better structural fitting, we updated our structural interpretation according to the suggestions from reviewers. In the revision (Figures 3, 6 and 7; Figure 3—figure supplements 1-4; Figure 7—figure supplement 1), the building block of S-OPA1 assembly in both nucleotide-free and GTP-γ-S bound states has been assigned as a new dimer, which shares the similar interface 1 and interface 2 in comparison with S-Mgm1 and Dyn1. However, in the nucleotide-free state, the unique interfaces I3 at the stalk region and P1 at the GMB region are also observed and described. These two unique interfaces disappear after GTP-γ-S binding with a new interface P2 of GMB region appearing, indicating a large assembly change of S-OPA1 after nucleotide binding. In the revision, the assemblies of S-OPA1 on membrane in both nucleotide-free and GTP-γ-S bound states are all left-handed (Figure 3—figure supplement 2). With this new structural interpretation, the model of how the assembly of S-OPA1 responses to GTP-γ-S binding is also revised (see Figure 7E). Besides, the comparison of oligomer assembly between S-OPA1, S-Mgm1 and Dyn1 are also re-performed (Figure 3—figure supplement 4).

2) As now the structures of the yeast Opa1 homolog Mgm1 are available, the authors are suggested to use the Mgm1 model to fit the cryo-EM density and make corresponding analysis. Considering the substantial difference between the Mgm1 paddle domain and dynamin PH domain, as well as the special kink of the Mgm1 stalk domain, more accurate models could be obtained by this analysis. In this case, the authors may add also the Mgm1 sequence into the alignment in Supplementary Figure 5A.

We would like to thank the reviewers for this important suggestion. In the revision, we have utilized the recent published crystal structure of S-Mgm1 (PDB code 6QL4) to re-perform structural fitting of both nucleotide-free and GTP-γ-S bound maps. In addition, the sequence alignment between S-OPA1 and S-Mgm1 was also performed (Figure 3—figure supplement 1).

3) As the structural difference between the nucleotide-free and GTP-γ-S bound states of Opa1 polymer is a major finding of this manuscript, the author should perform an ITC analysis to confirm and quantify the association between Opa1 and GTP-γ-S.

We would like to thank the reviewers for this important suggestion. Yes, we tried to perform ITC analysis to confirm GTP-γ-S binding. However, we found a severe protein precipitation during GTP or GTP-analogs titration, which stopped the possibility of using ITC to quantify the association between SOPA1 and GTP-γ-S.

As suggested below by reviewers, we then utilized different mutations of S-OPA1 at its GTPase domain to examine whether it is the GTP binding or hydrolysis to induce assembly change and thus tube expansion (Figure 4C). For those mutants that lose GTP hydrolysis activity but keep the ability of GTP binding, we observed the tube expansion effect after adding GTP. This further confirms it is the GTP binding to trigger the assembly change of SOPA1 on membrane.

4) Supplementary Figure 5: GMB comparison with PH domain in dynamin. A sequence alignment of dynamin and OPA1 shows a very low amino acid identity in the region of the PH and GMB domain. A reliable alignment is not possible in this region. The authors speculate that amino acids 794-800 form an amphipathic helix which may bind to the membrane. However, the envisaged sequence is less than two turns in length, which would constitute only a small membrane anchoring point. Second, the proposed OPA1 amino acids are aligned to an amino acid stretch in the PH domain of dynamin, which forms two β-strands (compare sequence alignment to the structure of the PH domain, Ferguson et al., 1994, pdb 2dyn). In fact, the aligned β-strand region in the PH domain contains a known membrane binding loop. These observations do not support the conclusion that membrane binding in OPA1 is mediated by an amphipathic helix. The investigation of charged and hydrophobic residues in the GMB domain in order to detect the membrane binding region is sufficient to explain the experiments in Figure 5 without invoking an amphipathic helix.

We agree with the reviewers. The amphipathic helix assumption and statement have been removed in the revision and we only discussed the characteristic charged and hydrophobic residues of 794-800 region in the GMB domain.

5) Figures 1E and 4C: It should be clearly stated which nucleotide was used in which experiment and whether GTP-γ-S-induced structural changes were visible in negative stain EM. Representative negative-stain images should be provided for both conditions. The experiments with G interface mutants in Figure 4C make only sense in the presence of GTP-γ-S, since no structural changes can be expected for these mutants in the absence of nucleotide.

We would like to thank the reviewers for this nice suggestion. In the original Figures 1E and 4C, we did not add nucleotide in liposome tubulation experiments. As suggested by reviewers, we further utilized negative stain EM to examine the potential structural changes induced after the addition of GTP for various G domain mutants. The new results have been added into Figure 4C. For those mutants that lose GTP hydrolysis activity but keep the ability of GTP binding, we observed the tube expansion effect after adding GTP. This further confirms it is the GTP binding to trigger the assembly change of SOPA1 on membrane.

[Editors' note: further revisions were suggested prior to acceptance, as described below.]

Essential revisions:1) The assembly of the stalks in the nucleotide-free structure is still confusing. The new fittings, depiction of stalk interfaces and the schematic Figure 7E show that both the nucleotide-free and GTP-γ-S bound sOPA1 use interface-1 and 2 to form a filament. These interfaces are also used in Mgm1 and have been shown by mutagenesis to contribute to assembly. In contrast, the newly observed interface-3 is unique to the nucleotide-free form, is opposite of all previously known stalk interfaces, and due to missing mutagenesis, its significance for assembly remains unknown. To avoid confusion, the authors should use interface-1 and 2 to define the OPA1 filament also in the nucleotide-free form, e.g. in Figure 3A and E, Figure 3—figure supplement 2 and Figure 7E left. In Figure 7E left, not the blue and green dimers, but the blue striped and blue dimers should form the filament, as in Figure 7E right. In this case, I1 and I2 are present within a filament, and I3 is in between filaments. It would be helpful for the reader if one such turn is shown in a unique color in Figure 2A or Figure 3A (similar as it is now with the 'I3 filament') to indicate whether such filament is right- or left-handed. How do the G domains in this case connect? One might expect that the putative G domain interface is then in between different rungs.

We appreciate the reviewers so much for this additional important suggestion. We tried to re-define S-OPA1 filament using the common interface-1 and 2 (see revised Figure 7E and Figure 3—figure supplement 2D and E). With this new definition, as the reviewers expected, the G domain dimerization interface in this case is now in between different rungs (see revised Figure 3—figure supplement 5). However, with this new definition, we found the filament becomes right-handed, which generates a new inconsistency with the left-handed assembly at GTP-γ-S bound state. Therefore, we would like to keep our original definition in the main figures but put this alternative one at the supplements.

Besides, we agree with the reviewers that the newly observed interface-3 is unique and its significance for filament assembly is not validated due to lack of mutagenesis. However, based on the fitted model, we calculated the areas of those interfaces, which yielded 1324 Å2 for interface-1, 124 Å2 for interface-2, and 1599 Å2 for interface-3. The area of interface-3 is large enough to play an important role to stabilize the assembly of S-OPA1. Thus, although the future mutagenesis work is needed, the role of interface-3 should not be neglected. To be noted, the computed area of interface-2 might not be accurate due to the existence of unfitted density there (see Figure 3D).

2) Figure 4C: The new data on the GTP-bound sOPA1 mutants are nicely contributing to the manuscript. However, tube diameters have to be quantified for the mutants in the absence and presence of GTP (for example as in Figure 6—figure supplement 1 or as average diameter) to make a convincing claim. It also appears from Figure 1F that the representative tubes of the Δ196-252 variant are somehow smaller compared to sOPA1. Could the authors provide also some quantification of these?

We would like to thank the reviewers very much for this good suggestion. The diameters of sOPA1 mutants coated tubes before and after adding GTP have been quantified with statistics (see Figure 5—figure supplement 2 in the revision), which is in consistency with our previous observation. The relevant text has been revised with this additional information.

Upon the reviewers’ request, we also added the diameter statistics of Δ196-252 coated tube at the nucleotide-free state (see Figure 2—figure supplement 1A in the revision). As the reviewers observed, the averaged diameter of Δ196-252 coated tube is a bit smaller than the wild type.

3) Nomenclature GMB domain: The fitted paddle is anything but a globular domain (unlike the previously assumed PH domain). Maybe 'Extended Membrane Binding Domain' (EMB) instead?

Corrected as the suggestion.

4) Figure 1D legend should contain the used GTP concentration. It appears from Figure 4B that the stimulation of wt is about 10-fold, not 70-fold?

The legend of Figure 1 has been revised accordingly. For the fold of activity stimulation by adding liposome, the parameter Kcat was used in this study for comparison (see Supplementary file 1 and Supplementary file 1—source data 1). The apparent increase of the activity can be represented by Kcat/Km, which is about 10-fold as mentioned by the reviewers.